# Patient Ability to Use Old versus New/Modified Model Adrenaline Autoinjection Emergency Medical Devices for Anaphylaxis in Prehospital Setting: A Systematic Review and Meta-Analysis

**DOI:** 10.3390/healthcare10020183

**Published:** 2022-01-18

**Authors:** Leong Chow Wei, Mohd Boniami Yazid, Mohd Noor Norhayati, Abu Yazid Md Noh, Andey Rahman

**Affiliations:** 1Department of Emergency Medicine, School of Medical Sciences, Universiti Sains Malaysia, Kubang Kerian, Kota Bharu 16150, Kelantan, Malaysia; lcwei28@hotmail.com (L.C.W.); abuyazid@usm.my (A.Y.M.N.); andey@usm.my (A.R.); 2Department of Family Medicine, School of Medical Sciences, Universiti Sains Malaysia, Kubang Kerian, Kota Bharu 16150, Kelantan, Malaysia; hayatikk@usm.my

**Keywords:** AAI, anaphylaxis, adrenaline

## Abstract

Background: The goal of this study was to determine the individual’s ability to use new/modified model AAI compared to old model AAIs devices for anaphylaxis. Methods: The protocol was established a priori and published on PROSPERO (CRD42021229691) and was conducted based on PRISMA guidelines. MEDLINE and CENTRAL were searched until 31 January 2021. Only RCTs were included in this review. Primary studies comparing old model AAI to new/modified model AAI emergency medical devices were included. Primary outcomes included number of successful administrations, and number of individuals to complete all steps. Secondary outcomes included successful removal of device safety guards, placement of correct end of the device against the thigh and holding of the device in place for adequate time after administration; the frequency of an adverse event (digital injection); individual preferences in terms of size, individual preference in terms of ease for carrying, overall patient preference; and the mean time of delivery. Results: Overall, seven trials consisting of 1359 patients were analyzed. Reporting of adverse events was limited to digital injection, which was significantly higher in the old model AAI (RR 6.90, 95% CI 3.27 to 14.57; I2 statistic = 0%; *p* < 0.001; four trials, 610 participants; high quality evidence). No significant difference was found regarding successful administration between the old model AAI and new/modified model AAI (RR 0.76, 95% CI 0.52 to 1.11; I2 statistic = 96%; *p* = 0.16; seven trials, 2196 participants; low quality evidence). Conclusions: We cannot make any new recommendations on the effectiveness of different models of AAIs regarding successful administration. However, considering the aspect of safety, we think that mew/modified model AAI can be chosen as the old model AAI was associated with a higher frequency of the adverse event (digital injection).

## 1. Introduction

### 1.1. Background

Anaphylaxis is the most dramatic and severe form of hypersensitivity reaction. Due to its rapid onset in nature, failure to diagnose and failure to give appropriate treatment in time may result in death [1,2].

Adrenaline (epinephrine) is the first and most important treatment for anaphylaxis, and it should be administered as soon as anaphylaxis is recognized to prevent the progression to life-threatening symptoms [3]. An adrenaline autoinjector (epinephrine autoinjector, AAI) is a medical device for injecting a measured dose or doses of adrenaline using autoinjector technology. The adrenaline delivered by the device is an emergency treatment for anaphylactic reaction [4]. Several types of adrenaline autoinjectors are available. These include the EpiPen (most used worldwide), the Anapen, the Adrenaclick, the Jext, and the Twinject. The first modern epinephrine autoinjector, the EpiPen, was invented in the mid-1970s. Although many years have passed, there is still a limited number of AAI currently available. As of 2018, three types of AAI were available in the US: Adrenaclick, Auvi-Q, and EpiPen. As of 2018, EpiPen is the only epinephrine autoinjector available for sale in Canada [5]. Unfortunately, each has limitations concerning dose, needle length, shelf life, and a lack of user-friendly design [6,7].

There are some limitations or adverse effects from using adrenaline autoinjectors. Physicians and other healthcare professionals must be trained (and re-trained regularly) to use the autoinjectors correctly and safely [8,9,10]. The ability to inject adrenaline correctly and safely through an autoinjector is not intuitive. Adverse effects such as injuries after unintentional injection of the adrenaline into fingers, thumbs, and other body parts have been reported [11,12].

### 1.2. Importance

Failure to use an adrenaline autoinjector to treat an anaphylaxis episode in the community has been reported [13]. Although AAI have limitations and uncertainties concerning their use, some reviews show that newer models of AAI may slightly increase the proportion of people correctly using the devices and reduce time to administer [14]. Therefore, this systemic review aims to determine individual ability to use old versus new/modified model autoinjection emergency medical devices. By demonstrating a difference in terms of use-competence between old and new model AAIs, we establish a reason to consider which AAI device is best prescribed for patients.

## 2. Methods

Our systematic review was conducted according to the protocol previously published in the PROSPERO registry: (www.crd.york.ac.uk/PROSPERO) (accessed on 1 December 2019). The protocol was established a priori and published on PROSPERO (CRD42021229691). The methodology and reporting were based on recommendations from the PRISMA guidelines, the Cochrane Collaboration and the preferred reporting items for systematic reviews and meta-analyses statement. It was evaluated according to the GRADE (grading of recommendations assessment, development, and evaluation) guidelines.

We searched the Cochrane Central Register of Controlled Trials CENTRAL (January 2021) and MEDLINE (1966 to January 2021). We used the search strategy in Appendix A to search MEDLINE and CENTRAL. We did not impose any language or publication restrictions. We searched for ongoing trials through the World Health Organization (WHO) International Clinical Trials Registry Platform (ICTRP) http://www.who.int/ictrp/en/ (accessed on 7 January 2021) and www.clinicaltrials.gov (accessed on 7 January 2021). We checked the reference list of identified RCTs and reviewed articles to find unpublished trials or trials not identified by electronic searches. We also contacted experts in the field and pharmaceutical companies which market AAI to identify unpublished trials.

We selected all randomized control trials (RCTs) comparing an old model AAI device with new/modified model AAI devices. We include blinded and open-label studies. We included individuals of all age, gender, and ethnicity with or without a history of anaphylaxis. In this review, our definition for old model AAI was EpiPen. Any model other than EpiPen, we referred to as modified/new model AAI devices. The intervention we selected was the old model AAI device, which refers to EpiPen. The comparison was modified/new model AAI devices for anaphylaxis, which refer to the Anapen, Auvi-Q, INT01, INT02, and TwinJect. The primary outcomes included multiple successful administrations, and a number of individuals able to complete all steps. Secondary outcomes included successful removal of device safety guards, proper placement of the correct end of the device against the thigh, and devices held in place for adequate time after administration. We further determined the frequency of one adverse event (digital injection), the overall preferences of patients, including individual preferences in terms of size, and ease of transport, and the mean time of delivery.

The planned subgroup analyses were age of individuals, (e.g., adult, or pediatric), and prior exposure to training. We were unable to carry out all subgroup analyses in the categories outlined in the protocol because there were insufficient data to be analyzed, as most of our RCT did not break down by age group.

Authors (LCW, and MBY) scanned the titles and abstracts independently from the searches and obtained full-text articles when they appeared to meet the eligibility criteria or where there was insufficient information to assess eligibility. We independently assessed the eligibility of the trials and documented the reasons for exclusion. We resolved any disagreements between the review authors (MNN, AY, and AR) by discussion. We contacted the trial authors if clarification was needed. 

None of the authors received any funding to eliminate a potential source of bias.

### 2.1. Data Collection and Processing

From each of the selected trials, we extracted study setting, participant characteristics (age, gender, ethnicity), methodology (number of participants randomized and analyzed), type of adrenalin autoinjection device used, successful administrations of the adrenaline autoinjection device, ability to complete all steps when using the adrenalin autoinjection device, successful removal of the device’s safety guard, placement of the correct end of the device against the thigh, if the device was held in place for adequate time after administration, frequency of adverse events (digital injection), overall preference of autoinjection device, preference in terms of size of autoinjection device, preference in terms of ease of carrying the autoinjection device, and mean time for delivery when using the autoinjection device. Three authors were involved in risk of bias assessment (LCW, MBY, and MNN). We assessed the risk of bias based on random sequence generation, allocation concealment, blinding of participants and personnel, blinding of the outcome assessors, completeness of the outcome data, selectivity of outcome reporting and other bias discussed in the *Cochrane Handbook for Systematic Reviews of Interventions* [15]. We resolved any disagreements by discussion.

### 2.2. Grading Quality of Evidence

We assessed the quality of evidence for all primary and secondary outcomes according to the GRADE methodology for assessing risk bias: inconsistency, indirectness, imprecision, and publication bias; classified as very low, low, moderate, or high.

### 2.3. Primary Data Analysis

We undertook meta-analyses using Review Manager 5.4 software (The Cochrane Collaboration St Albans House, 57–59 Haymarket, London SW1Y 4QX, United Kingdom) and used a random-effects model to pool data. Thresholds for the interpretation of the I^2^ statistic can be misleading, since the importance of inconsistency depends on several factors. We used the guide to interpret heterogeneity as outlined: 0% to 40% might not be important; 30% to 60% may represent moderate heterogeneity; 50% to 90% may represent substantial heterogeneity; and 75% to 100% would be considerable heterogeneity [15]. We assessed the presence of heterogeneity in two steps. First, we assessed obvious heterogeneity at face value by comparing populations, settings, interventions and outcomes. Second, we assessed statistical heterogeneity by means of the I2 statistics [15]. We drew forest plots for the trials with categorical outcomes using risk ratios (RR) and 95% confidence intervals (CI) and we also calculated risk differences (RD) and 95% CI. If we had encountered numerical outcomes, we intended to analyze these using mean differences (MD) and 95% CI. We checked included trials for unit of analysis errors, and we did not encounter any of these. If we had encountered any cluster-RCTs we intended to adjust the results from trials showing unit of analysis errors based on the mean cluster size and intracluster correlation coefficient [15]. We contacted the original trial authors to request missing or inadequately reported data. We performed analyses on the available data in case missing data were not available. We performed sensitivity analysis to investigate the impact of risk of bias for sequence generation and allocation concealment of included studies. When there were sufficient studies, we used funnel plots to assess the possibility of reporting biases, small study biases, or both.

## 3. Results

We retrieved 72 records by searching electronic databases and 22 records from other sources (Figure 1). We screened a total of 80 records. We excluded a total of 72 records, the main reason being analyses of those trials were irrelevant within our objective, such as the measurement of the bioavailability, pharmacokinetics and pharmacodynamics of adrenaline in anaphylaxis, evaluation of how communities manage patients with anaphylaxis, assessment of effective training and skill retention for the public use of AAIs, and assessment of the impact of anaphylaxis on health-related quality of life (HRQL). We reviewed full copies of eight studies: One study was excluded, as this study compared an old model AAI vs. pre-filled adrenaline syringes instead of new/modified model AAIs [16]. Therefore, we included seven trials and excluded one trial from the review. We included seven trials with a total of 1363 participants [17,18,19,20,21,22,23]. All seven trials claimed not to have received funding from AAI device manufacturers.

Six of the seven trials were conducted in high-income countries [18,19,20,21,22,23] and one trial in a middle-income country [17]. Three of the seven trials recruited participants from healthcare settings [18,21,23], two trials recruited participants from university settings [17,22], and two trials recruited participants from research/laboratory facilities [19,20]. Six trials reported exclusion of participants due to previous AAI knowledge or experience; either they were in poor health; were unable to read; were not native English speakers, they or another member of the household were employed by, affiliated with, or engaged in any organization that might introduce potential bias into the study; had experience with severe or life-threatening allergies (themselves or close family members); worked in a profession that may introduce bias were children weighing <7.5 kg; or were experiencing significant psychiatric problems such as psychotic disorders [17,18,19,20,21,23]. All trials involving 1363 participants mentioned the gender of the participants, and these were equally distributed throughout the seven trials and across the intervention and comparison groups [17,18,19,20,21,22,23]. Participants in the trials were randomly placed into intervention and control groups. For three trials, the participants involved were crossovers [19,20,22]. For five trials, an instruction sheet was provided to the participants regarding the use of the device [17,19,20,21,22]. Demonstration regarding individual use of the device was provided in two trials [18,23]. Three trials compared with the Anapen [18,21,23], two trials compared with the Auvi-Q [19,20], one trial compared with a modified EpiPen [17], and one trial compared with the TwinJect [22]. Four trials specifically reported that they had measured the number of successful administrations for EpiPen compared to other AAI devices and the number of individuals to complete all steps when using EpiPen compared with other AAI devices [18,20,21,23]. Another three studies measured the number of individuals who completed all steps using EpiPen compared with other AAI devices [17,19,22]. We included these three trials that measured the number of successful administrations for EpiPen compared to other AAI devices. It was reasonable to expect that individuals who completed all steps must be successfully administering the drug. One trial reported the outcome after a period of six weeks and one year of participant follow-up. We used the outcome assessment after sixth-week follow-up because it gave better consistency across the included trials [23]. Seven trials reported the secondary outcomes of successful removal of the safety guard, placement of the correct end of the device against the thigh and holding of device in place for an adequate time after administration [17,18,19,20,21,22,23]. The secondary outcome regarding the frequency of adverse digital injection between EpiPen and other AAI devices was reported by four trials [17,20,22,23]. Digital injection means accidental self-injection of the adrenaline pen into the finger. The secondary outcome regarding patients’ overall preference for EpiPen compared to other AAI devices was reported by three trials [19,20,22]. Individual preferences in terms of the size of EpiPen compared to other AAI devices and regarding ease of transport between EpiPen and other AAI devices were reported in two trials [19,22], and he mean time of delivery was reported by three trials [17,19,20]. The characteristics of included trials are shown in Table 1.

### 3.1. Risk of Bias in Included Studies

Overall, four randomized control trials were categorized as exhibiting low risk of bias [19,20,22,23] and three were categorized as unclear [17,18,21]. Figure 2 and Figure 3 summarize the risks of bias.

### 3.2. Allocation

Four trials described the method of randomization used. One trial randomized the participants according to randomization schedules generated [19], and three trials used computer-generated randomization [18,21,23]. The randomization method was not reported in the other three trials, and, thus, we judged random sequence generation as an unclear risk of bias [17,20,22]. Allocation concealment was unclear in three trials [17,18,21].

### 3.3. Blinding

One trial used grey adhesive paper to conceal the instruments [17]. Two trials did not allow participants to access any product leaflets or independently determine how to use the AAI by relying on written or voiced instructions on the device label and/or device [19,20]. One trial stated that the participants were not allowed to view the devices before beginning the study [22]. Blinding of participants and personnel was not described in three trials [18,21,23].

### 3.4. Incomplete Outcome Data

Seven trials measured the primary and secondary outcomes and were included in the meta-analysis. Of these, one trial measured the outcomes at six weeks and had less than a 20% loss during follow-up [23].

### 3.5. Selective Reporting

Seven trials reported the outcomes as specified in their methods section [17,18,19,20,21,22,23].

### 3.6. Other Potential Sources of Bias

We detected no other potential sources of bias.

### 3.7. Primary Outcomes

Seven trials reported more successful administrations for old model AAIs compared to other AAI devices [17,18,19,20,21,22,23]. There was no significant difference between the old model AAI and the control group (RR 0.76, 95% CI 0.52 to 1.11; I2 statistic = 96%; *p* = 0.16; seven trials, 2196 participants; low quality evidence) (Figure 4, Table 2).

Seven trials documented the number of individuals who completed all steps [17,18,19,20,21,22,23]. The number of individuals who completed all steps was lower in the old model AAI group compared to the group using the new/modified model AAI device (RR 0.57, 95% CI 0.33 to 0.97; I2 statistic = 95%; *p* = 0.04; seven trials, 2196 participants; low quality evidence) (Figure 5, Table 2).

### 3.8. Secondary Outcomes

Seven trials reported successful removal of the safety guard [17,18,19,20,21,22,23]. Old model AAIs showed no significant difference in outcome of success at removing the safety guard compared to the control (RR 0.98, 95% CI 0.91 to 1.06; I2 statistic = 88%; *p* = 0.64; seven trials, 2196 participants; low quality evidence) (Figure 6, Table 2).

Seven trials reported placement of the correct end of the device against the thigh [17,18,19,20,21,22,23]. In the old model AAI group, there was a significant reduction in the number of this outcome (RR 0.75, 95% CI 0.62 to 0.91; I2 statistic = 95%; *p* = 0.003; seven trials, 2196 participants; moderate quality evidence) (Figure 7, Table 2).

Seven trials reported data regarding whether the device was held in place for adequate time after administration [17,18,19,20,21,22,23]. There was no significant difference between the old model AAI and control groups (RR 0.77, 95% CI 0.59 to 1.01; I2 statistic = 94%; *p* = 0.06; seven trials, 2196 participants; moderate quality evidence) (Figure 8, Table 2).

Four trials reported a significant increase in the frequency of the adverse event (digital injection) in the old model AAI group [17,20,22,23] (RR 6.90, 95% CI 3.27 to 14.57; I2 statistic = 0%; *p* < 0.001; four trials, 610 participants; high quality evidence) (Figure 9, Table 2).

Two trials reported significant reduction in patient overall preference of old model AAI compared to other AAI devices with substantial heterogeneity [19,20] (RR 0.12, 95% CI 0.05 to 0.29; I2 statistic = 82%; *p* < 0.001; two trials, 1578 participants; high quality evidence) (Figure 10, Table 2).

One trial reported the individual preference in terms of size [19]. This trial showed significant reduction in individual preference in terms of size for the old model AAI group. One trial reported the individual preference in terms of ease of carrying, showing significant reduction in individual preference for carrying the old model AAI devices [19]. Three trials reported the mean time of delivery but without a measurement of standard deviation, we were not able to analyze this outcome [17,19,20]. Those trials showed increases in the meantime of delivery were experienced by the old model AAI group.

## 4. Discussion

This review was designed to include all RCTs addressing the ability of patients to use old model versus new/modified model AAI devices for anaphylaxis. The results suggest that there was not a significant reduction in successful administrations (primary outcome), but the number of individuals who completed all steps was significantly reduced in the old model AAI. It seems like whether people completed all steps or not, they were just as likely to administer successfully regardless of the device. However, when we screened through the RCT, we unable to find out which critical step most often led to successful administration. 

Reporting of adverse events was limited to digital injection, which significantly increased with use of the old model AAI, but we unable to proceed with subgroup analysis due to limited data in each RCT. Therefore, it is uncertain whether this differed depending on people’s age or other characteristics such as whether they were trained to administer injections or whether they had a food allergy. The number of people in the studies used in this outcome was also relatively small.

There was a significant reduction in the outcome of placement of the correct end of device against the thigh and in the outcome of overall patient preference of the old model AAI. The patient overall preference here refers to preference in terms of size, shape, ease of carrying, ease of administration, clarity of device instruction, durability, confident use during emergency, and safety.

There was no significant difference in the number of successful administrations, successful removal of safety guards, or proper holding of the device in place for an adequate time after administration between the old model AAI and the control group. 

We were not able to analyze the delivery time between old versus new/modified model AAIs. Three of the seven trials assessing the outcome of mean time of delivery were not included, because neither standard deviation or *p*-value were mentioned. Therefore, this limits the applicability of the findings in this review.

The quality of evidence was low to high. Generally, there was a low or unclear risk of bias for most trials in most domains. For the outcome of the frequency of the adverse event of digital injection, and the outcome of patient overall preference of old model AAI, the quality of the evidence was judged to be high. For the outcome of holding the device in place for adequate time after administration, and the outcome of successful removal of the safety guard, the certainty of evidence was moderate. The certainty of the evidence for the outcome of number of successful administrations, the outcome of number of individuals to complete all steps, and the outcome of placement of correct end of the device against the thigh was low. The way that certainty of evidence was calculated is shown in Table 2.

### 4.1. Relation to Other Reviews

To our knowledge, there are no other reviews regarding patient ability to use old versus new/modified model AAI devices for anaphylaxis. We found a review regarding adrenaline autoinjectors for the treatment of anaphylaxis with and without cardiovascular collapse in the community [24]. However, the objective of the review only assessed the effectiveness of adrenaline (epinephrine) auto-injectors in relieving respiratory, cardiovascular, and other symptoms during episodes of anaphylaxis that occur in the community instead of comparing the model of AAI devices.

### 4.2. Limitations

This systemic review and meta-analysis has several limitations. First, we attempted to reduce publication bias by checking the reference lists of all related studies for further references and searched multiple databases without language restriction. However, we cannot be certain that we have located all the trials in this area. Although there were seven included trials, not all included trials reported all outcomes. We also encountered considerable heterogeneity in our primary outcome. We were not able to explain this in our subgroup analysis due to data limitations in each RCT provided. The biggest contribution to heterogeneity was probably using different comparators. Besides that, most of the studies did not use standardized steps to define successful administration and complete all steps. Different sample sizes in each RCT, plus drawing participants from different settings, particularly from lab settings also contributed to the bias and heterogeneity of our systemic review and meta-analysis. 

## 5. Conclusions

Based on this review, we cannot make any new recommendations on the effectiveness of different models of AAI regarding successful administration for the treatment of anaphylaxis due to low certainty of evidence. When considering the aspect of safety, we think that mew/modified model AAIs can be chosen, as the old model AAI was associated with high frequency of adverse event (digital injection). From the point of view of the patient, the new/modified model AAI has a high overall preference. However, cost of the device was not included in our consideration of overall preference. Future studies can include the cost of each device, so that the financial aspect can be considered when choosing the best device for patients.

Our study lacks information regarding the level of patient training or prior exposure to AAI devices. We are uncertain if this may alter the usability for those AAI devices. 

Future studies addressing this research question might benefit by focusing on some of the limitations that we encountered with current evidence. To reduce heterogeneity, standardized steps/instructions to use the device should be developed and used in future research. Future research should provide a more precise exploration of subgroup samples in terms of age, previous training, and the education level of patients.

## Figures and Tables

**Figure 1 healthcare-10-00183-f001:**
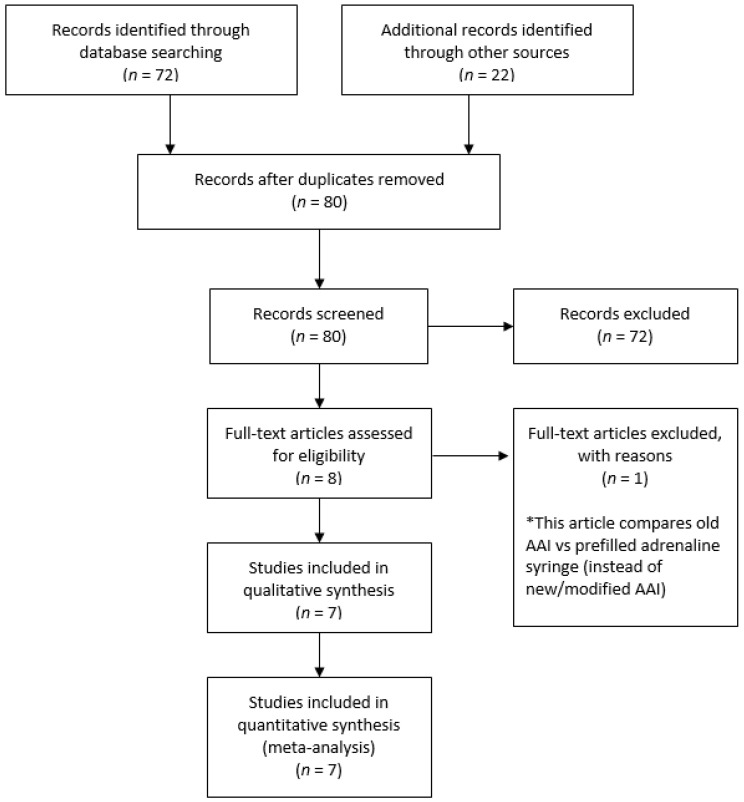
Study flow diagram.

**Figure 2 healthcare-10-00183-f002:**
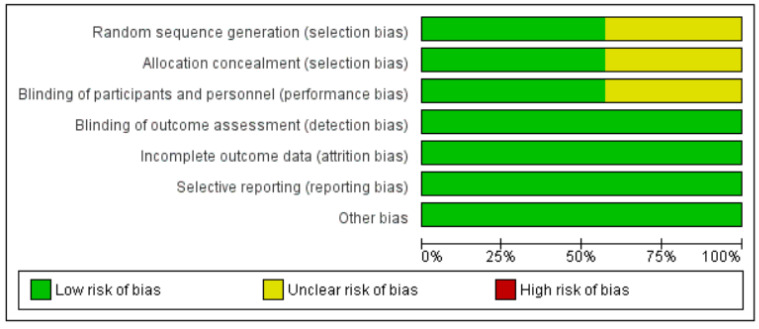
‘Risk of bias’ graph: review authors’ judgements about each risk of bias item presented as percentages across all included studies.

**Figure 3 healthcare-10-00183-f003:**
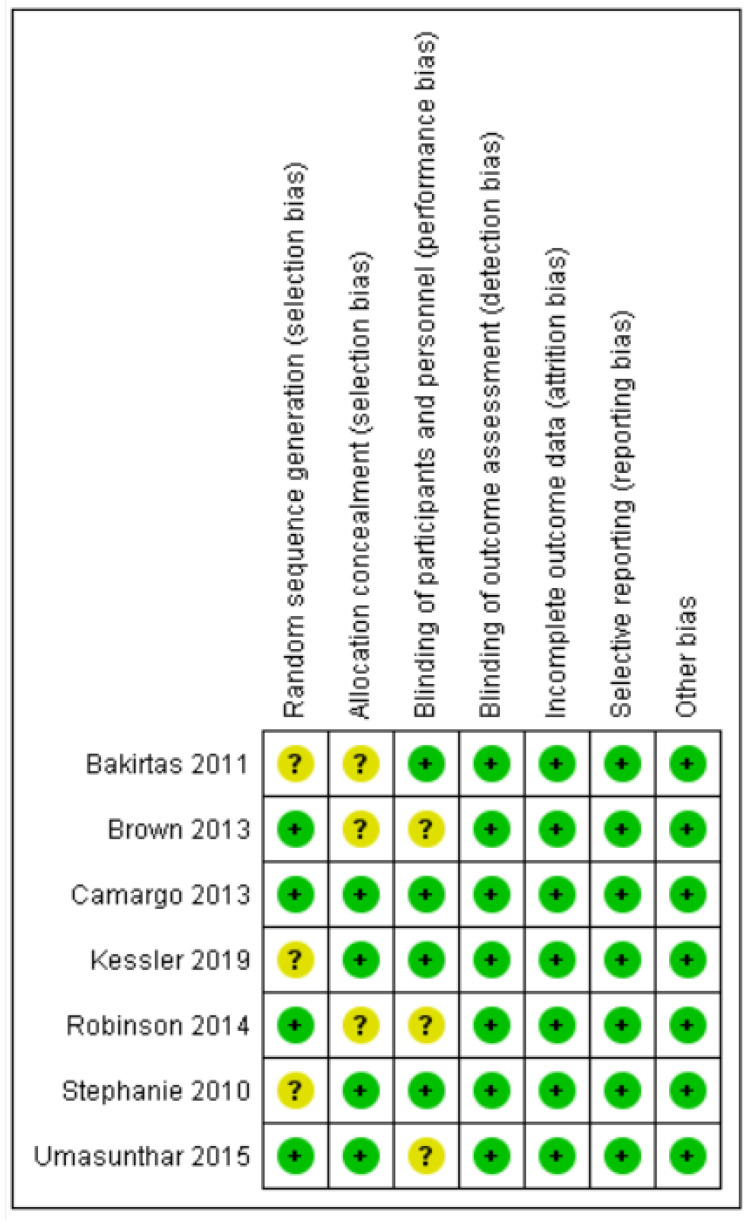
‘Risk of bias’ summary: review authors’ judgements about each risk of bias item for each included study.

**Figure 4 healthcare-10-00183-f004:**
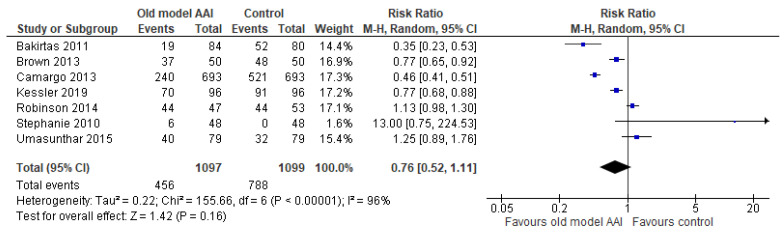
Forest plots for the outcome number of successful administrations.

**Figure 5 healthcare-10-00183-f005:**
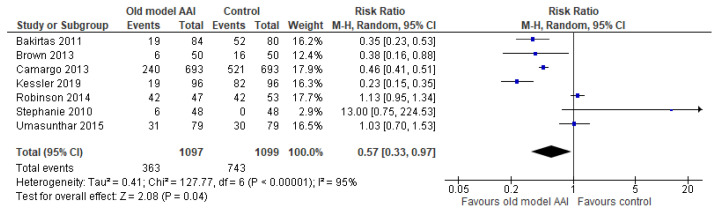
Forest plots for the outcome number of individuals who completed all steps.

**Figure 6 healthcare-10-00183-f006:**
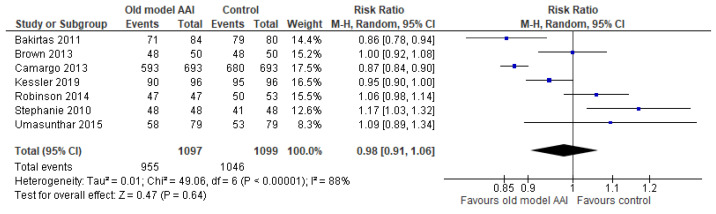
Forest plots for the outcome successful at remove the safety guard.

**Figure 7 healthcare-10-00183-f007:**
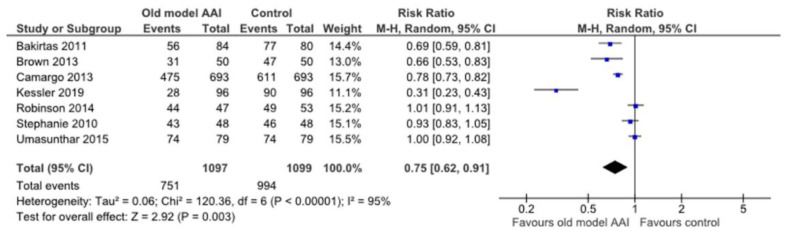
Forest plots for the outcome placement of correct end of the device against the thigh.

**Figure 8 healthcare-10-00183-f008:**
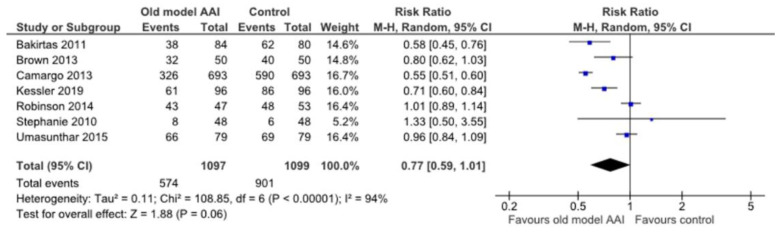
Forest plots for the outcome hold devices in place for adequate of time after administration.

**Figure 9 healthcare-10-00183-f009:**
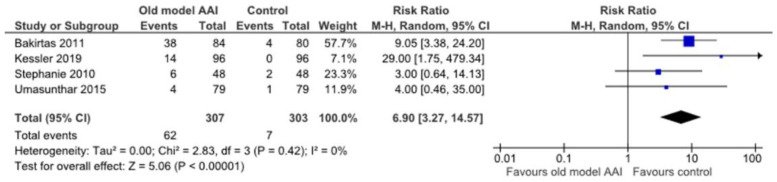
Forest plots for the outcome the frequency of adverse event (digital injection).

**Figure 10 healthcare-10-00183-f010:**
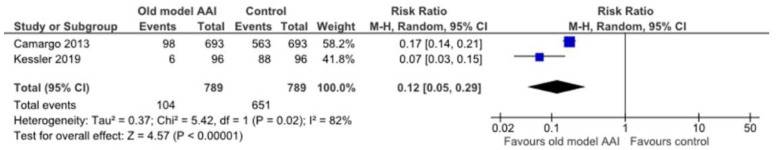
Forest plots for the outcome patient overall preference.

**Table 1 healthcare-10-00183-t001:** Characteristics of included trials.

Reference	Country	No of Patients/No of Trial Sites	Clinical Setting	Intervention	Type of Control	Funding from AAI Device Manufacturers	Age of Participants
Bakirtas 2011 [17]	Turkey	164/1	All interns of 2009–2010 terms in Gazi University Faculty of Medicine	Epipen	Modified Epipen	No	NA
Brown 2013 [18]	United Kingdom	100/1	Mothers in general pediatric outpatient departments and inpatient children’s wards	Epipen	Anapen	No	NA
Camargo 2013 [19]	United States	693/12	Individuals in research facilities	Epipen	Auvi-Q	No	11–65 years
Kessler 2019 [20]	United States	96/1	Individuals in laboratory in Bala Cynwyd	Epipen Jr	Auvi-Q	No	18–65 years
Robinson 2014 [21]	Australia	100/1	Visitors, parents and hospital staff of Royal Children’s Hospital	Epipen	Anapen	No	NA
Stephanie 2010 [22]	Virginia	48/1	Native English speakers in the Community of Charlottesville	Epipen	TwinJec	No	7–55 years
Umasunthar 2015 [23]	United Kingdom	158/1	Mothers of children aged 0–18 in specialist pediatric allergy center	Epipen	Anapen	No	NA

**Table 2 healthcare-10-00183-t002:** Summary of findings for the main comparison.

Patient Ability to Use Old versus New/Modified Model Auto-Injection Emergency Medical Devices for Anaphylaxis in Pre-Hospital Setting						
Patient or Population: Adult and Children Setting: University, Hospital and Research Facilities Intervention: Old Model AAI Comparison: New/Modified Model AAI						
Outcomes	Anticipated Absolute Effects * (95% CI)		Relative Effect (95% CI)	№ (s)	Certainty	Quality Assessment Domains
	Risk with New/ Modified Model Auto Injection Device	Risk with Old Model Auto Adrenaline Injection				
Number of successful administrations	717 per 1000	**545 per 1000** (373 to 796)	**RR 0.76** (0.52 to 1.11)	2196 (7 RCTs)	⨁⨁◯◯ LOW	Risk of bias: Not serious * Inconsistency: Serious † Indirectness: Not serious Imprecision: Serious ¶ Publication bias: None Large effect: No
Number of individuals to complete all steps	676 per 1000	**385 per 1000** (223 to 656)	**RR 0.57** (0.33 to 0.97)	2196 (7 RCTs)	⨁⨁◯◯ LOW	Risk of bias: Not serious * Inconsistency: Serious †† Indirectness: Not serious Imprecision: Serious ¶¶ Publication bias: None Large effect: No
Successful to remove safety guard	952 per 1000	**933 per 1000** (866 to 1000)	**RR 0.98** (0.91 to 1.06)	2196 (7 RCTs)	⨁⨁◯◯LOW	Risk of bias: Not serious * Inconsistency: Serious ††† Indirectness: Not serious Imprecision: Serious ¶¶¶ Publication bias: None Large effect: No
Placement of correct end of the device against the thigh	904 per 1000	**678 per 1000** (561 to 823)	**RR 0.75** (0.62 to 0.91)	2196 (7 RCTs)	⨁⨁⨁◯ MODERATE	Risk of bias: Not serious * Inconsistency: Serious †††† Indirectness: Not serious Imprecision: Not serious ¶¶¶¶ Publication bias: None Large effect: No
Hold device in place for adequate of time after administration	820 per 1000	**631 per 1000** (484 to 828)	**RR 0.77** (0.59 to 1.01)	2196 (7 RCTs)	⨁⨁⨁◯ MODERATE	Risk of bias: Not serious * Inconsistency: Serious ††††† Indirectness: Not serious Imprecision: Not serious ¶¶¶¶¶ Publication bias: None Large effect: No
The frequency of adverse event (digital injection)	23 per 1000	**159 per 1000** (76 to 337)	**RR 6.90** (3.27 to 14.57)	610 (4 RCTs)	⨁⨁⨁⨁ HIGH	Risk of bias: Not serious ** Inconsistency: Not serious § Indirectness: Not serious Imprecision: Not serious ‡ Publication bias: None Large effect: Very large
Patient overall preference	825 per 1000	**99 per 1000** (41 to 239)	**RR 0.12** (0.05 to 0.29)	1578 (2 RCTs)	⨁⨁⨁⨁ HIGH	Risk of bias: Not serious *** Inconsistency: Serious §§ Indirectness: Not serious Imprecision: Not serious ‡‡ Publication bias: None Large effect: Large
* **The risk in the intervention group** (and its 95% confidence interval) is based on the assumed risk in the comparison group and the **relative effect** of the intervention (and its 95% CI). **CI:** Confidence interval; **RR:** Risk ratio						
**GRADE Working Group grades of evidence** **High certainty:** We are very confident that the true effect lies close to that of the estimate of the effect **Moderate certainty:** We are moderately confident in the effect estimate: The true effect is likely to be close to the estimate of the effect, but there is a possibility that it is substantially different **Low certainty:** Our confidence in the effect estimate is limited: The true effect may be substantially different from the estimate of the effect **Very low certainty:** We have very little confidence in the effect estimate: The true effect is likely to be substantially different from the estimate of effect						

**Explanations**: * Overall, 4 lower and 3 unclear risks of bias trials. † I2 = 96%, *p* < 0.01 for heterogeneity. ¶ 3 of 7 trials showing increased number of successful administrations with old model AAI. †† I2 = 95%, *p* < 0.01 for heterogeneity. ¶¶ 3 of 7 trials showing increased number of individuals to complete all steps with old model AAI. ††† I2 = 88%, *p* < 0.01 for heterogeneity. ¶¶¶ 3 of 7 trials showing increased number of the outcome successful at remove the safety guard with old model AAI. †††† I2 = 95%, *p* < 0.01 for heterogeneity. ¶¶¶¶ 1 of 7 trials showing increased in the outcome placement of correct end of the device against the thigh with old model AAI. ††††† I2 = 94%, *p* < 0.01 for heterogeneity. ¶¶¶¶¶ 2 of 7 trials showing increased the outcome hold devices in place for adequate of time after administration with old model AAI. ** Overall, 3 lower and 1 unclear risks of bias trials. § I2 = 0%, *p* = 0.42 for heterogeneity. ‡ No trials show reduced in the outcome the frequency of adverse event (digital injection) with old model AAI. *** Overall, 2 lower risks of bias trials. §§ I2 = 82%, *p* = 0.02 for heterogeneity. ‡‡ No trials show increased in the outcome of patient overall preference with old model AAI.

## Data Availability

The data presented in this study are available within the article.

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
