# Peer review of "Patient Ability to Use Old versus New/Modified Model Adrenaline Autoinjection Emergency Medical Devices for Anaphylaxis in Prehospital Setting: A Systematic Review and Meta-Analysis"

_healthcare, 2022, doi:10.3390/healthcare10020183_

Round 1

Reviewer 1 Report

Summary

The authors have undertaken a systemic review of the literature on the usability of newer AAI devices (correct administration, patient preference and adverse reactions) compared to the origin Epipen design (available in most countries until ~ 2010-11). 7 of 80 detected studies were included; none were funded by the manufacturers. Conclusions were that usability was similar for most devices examined, but that newer devices might be preferable due to lower rates of accidental needle-stick injuries. The newer devices examined were the modified Epipen, Anapen, Auvi-Q and TwinJec,

Results

The authors state that 1 study was excluded as it did not compare two different AAI; can the authors clarify why the many other studies were excluded?

Overall, correct placement against the thigh was significantly higher for the newer devices than the old Epipen (moderate quality evidence), needle stick injuries were also lower for the newer devices (high quality evidence) and patient preference was for the newer devices (high quality evidence).

Comments

There are a number of relevant issues in the use of AAI to treat anaphylaxis and a large literature on the same: compliance with carrying an AAI for use, whether the device is in date, whether it is used during anaphylaxis even when indicated and correctness of administration techniques (with implications regarding efficacy and accidental needle stick injuries. The authors have examined the latter two issues. The takeaway message is that newer devices are preferable to the old Epipen device design due to greater usability and less needle stick injuries with newer devices. The implication is that AAI using the old Epipen design (and similar generics) are best avoided. My only comment here is that the number of studies is relatively small and as we have no information regarding patient training, there may be other reasons for the authors conclusions.

Author Response

Thank for the review.

Reviewer 2 Report

General comment

The paper is quite interesting and I would like to congratulate the authors.

In order to improve the manuscript I address the following criticisms

Methodologic concern:

In the title the authors define the present study a systematic review and a meta-analysis. In the text the authors define it alternatively meta-analysis (es: line 313) or review (es. line 325). For the methodological approach the paper seems to be a meta-analysis. In this case the clinical setting of seven analyzed studies seem too different to combine.

Minor concerns:

  • Line 152: Specify briefly the reasons of 72 records exclusion
  • Line 157: Authors said “All seven trials declared funding from AAI device manufactures” but table 1 indicated no funding from them
  • Table 1: Add references number
  • Figure 2-3: Referred to Cochrane Collaboration’s Tool into the figures legend. Please for each paper carefully specify the main bias also into the text.
  • Lines 214: Please precise “no significant difference”
  • Lines 242-248: The sentences are quite unclear. Are one or two studies reporting the individual preference in term of size? This sentence (line 242-244) is doubled with different references (reference n°19 and n°34). Please provide more details about it.

Author Response

Thank for the review.
